# A Transcriptomic and Metabolomic Study on the Biosynthesis of Iridoids in *Phlomoides rotata* from the Qinghai–Tibet Plateau

**DOI:** 10.3390/plants13121627

**Published:** 2024-06-12

**Authors:** Luhao Wang, Guigong Geng, Huichun Xie, Lianyu Zhou, Yujiao He, Zuxia Li, Feng Qiao

**Affiliations:** 1Key Laboratory of Tibetan Plateau Medicinal Plant and Animal Resources, School of Life Sciences, Qinghai Normal University, Xining 810008, China; wangluhao188@163.com (L.W.); yezino.1@163.com (H.X.); zly7604@163.com (L.Z.); heyj108@163.com (Y.H.); 18909789774@163.com (Z.L.); 2Academy of Agricultural and Forestry Sciences, Qinghai University, Xining 810016, China; genggg-298@163.com; 3Academy of Plateau Science and Sustainability, Qinghai Normal University, Xining 810008, China

**Keywords:** *Phlomoides rotata*, transcriptomics, metabolomics, qRT-PCR, iridoid synthesis pathway

## Abstract

*Phlomoides rotata* is a traditional Chinese herbal medicine that grows in the Qinghai–Tibet Plateau region at a 3100–5000 m altitude. Iridoid compounds are the main active compounds of the *P. rotata* used as medical ingredients and display anti-inflammatory, analgesic, and hepatoprotective properties. To better understand the biological mechanisms of iridoid compounds in this species, we performed a comprehensive analysis of the transcriptome and metabolome of *P. rotata* leaves from four different regions (3540–4270 m). Global metabolome profiling detected 575 metabolites, and 455 differentially accumulated metabolites (DAMs) were detected in *P. rotata* leaves from the four regions. Eight major DAMs related to iridoid metabolism in *P. rotata* leaves were investigated: shanzhiside methyl ester, 8-epideoxyloganic acid, barlerin, shanzhiside, geniposide, agnuside, feretoside, and catalpin. In addition, five soil physical and chemical indicators in *P. rotata* rhizosphere soils were analyzed. Four significant positive correlations were observed between alkaline nitrogen and geniposide, exchangeable calcium and geniposide, available potassium and shanzhiside, and available phosphorus and shanzhiside methyl ester. The transcriptome data showed 12 *P. rotata* cDNA libraries with 74.46 Gb of clean data, which formed 29,833 unigenes. Moreover, 78.91% of the unigenes were annotated using the eight public databases. Forty-one candidate genes representing 23 enzymes involved in the biosynthesis of iridoid compounds were identified in *P. rotata* leaves. Moreover, the *DXS1*, *IDI1*, *8-HGO1*, and *G10H2* genes associated with iridoid biosynthesis were specifically expressed in *P. rotata*. The integration of transcriptome and metabolome analyses highlights the crucial role of soil physical and chemical indicators and major gene expression related to iridoid metabolism pathways in *P. rotata* from different areas. Our findings provide a theoretical foundation for exploring the molecular mechanisms underlying iridoid compound accumulation in *P. rotata*.

## 1. Introduction

Iridoids, a large class of cyclopentane pyran monoterpenes, are composed of two basic carbon frameworks: substituted iridoids and secoiridoids [1]. They are mostly in the form of glycosides and combine with glucose at the C-1 hydroxyl group. These metabolites, which are found in natural and traditional Chinese medicine, have been shown to have anti-inflammatory [2,3], hepatoprotective [4], liver protective [5,6], neuroprotective [7], and antioxidant activities [8]. The biosynthetic pathway for iridoid compounds can be divided into three stages: precursor formation, terpenoid skeleton biosynthesis, and subsequent chemical modification [9,10,11,12]. Precursor synthesis primarily involves the mevalonate pathway (MVA) and methylerythritol 4-phosphate pathway (MEP), which generate the common terpenoid precursors isopentenyl pyrophosphate (IPP) and dimethylallyl pyrophosphate (DMAPP). These precursors are then converted into the iridoid compound skeleton, iridodial, through catalysis by geranyl diphosphate synthase (GPPS), geraniol synthase (GES), geraniol 8-hydroxylase (G10H), and four other enzymes. The final iridoid compounds are formed through the chemical modification of 7-deoxyloganetic acid glucosyltransferase (7-DLGT), 7-deoxyloganic acid hydroxylase (7-DLH), and four other enzymes [13].

Gene expression plays a crucial role in regulating iridoid synthesis and accumulation in plants. The expression of specific genes related to iridoid biosynthesis in plants can regulate enzyme activity and affect iridoid biosynthesis [14,15,16]. A study on *Gentiana macrophylla* revealed a close relationship between the expression of *7-DLH* and secologanin synthase (*SLS*) genes and the production of iridoid compounds [17]. The expression of *7-DLH* and *SLS* was higher in flowers, indicating that flowers are important for iridoid biosynthesis in *G. macrophylla* [17]. Overexpression of the 4-hydroxy-3-methylbut-2-enyl diphosphate reductase (*HDR*) gene in *Salvia miltiorrhiza* has been shown to promote tanshinone synthesis in hairy roots [18]. Similarly, overexpression of the 1-deoxy-D-xylulose-5-phosphate synthase (*DXS*) gene in *Pinus massoniana* can enhance DXS enzyme activity, resulting in a significant increase in the carotenoid, chlorophyll a, and chlorophyll b content [19]. Furthermore, several studies have highlighted the role of transcription factor families such as *bHLH*, *MYB*, and *WRKY* in the plant secondary metabolism and the regulation of key genes in the iridoid synthesis pathway [20,21]. For example, overexpression of the *BIS3 (bHLH)* gene in *Catharanthus roseus* petals leads to the upregulation of loganic acid O-methyltransferase (*LAMT*) and *SLS* gene expression, consequently significantly increasing loganic acid accumulation [22].

*Phlomoides rotata*, a perennial herb belonging to the genus Phlomoides in the Lamiaceae family, is primarily distributed in meadows and hilly areas at altitudes ranging from 3100 m to 5000 m [23,24]. This unique medicinal plant on the Qinghai-Tibet Plateau is a traditional Chinese medicine widely used for the treatment of postoperative trauma, soft tissue injury, rheumatic bone pain, and other ailments [23,25,26]. Various compounds have been isolated from *P. rotata*, including flavonoids, iridoids, and phenylethanoid glycosides [26,27]. According to the research conducted by Zhan et al., the total flavonoids extracted from *P. rotata* demonstrated a significant anti-rheumatoid arthritis effect. Concurrently, the study identified luteoloside as the pivotal active compound contributing to this therapeutic impact [28]. The total polyphenolic glycoside extracted from *P. rotata* exhibits potential therapeutic efficacy in mitigating hepatic fibrosis [29,30]. Lei et al.’s research on *P. rotata* indicates that the total iridoid glycoside is instrumental in the transition from inflammation to cell proliferation [31]. Notably, iridoid compounds such as barlerin and shanzhiside methyl ester are the most important active ingredients in *P. rotata* and exhibit significant anti-inflammatory and analgesic effects [32,33,34]. Currently, research on *P. rotata* has predominantly focused on its distribution, extraction of active ingredients, and pharmacological mechanisms [23,35]. However, the iridoid distribution in different regions of *P. rotata* leaves and the expression of genes related to the iridoid pathway remain unclear. In this study, we explored the transcriptome and metabolome changes in the leaves of *P. rotata* from different regions using RNA-Seq sequencing and a wide-targeted metabolomics approach. Second, the physical and chemical indicators of *P. rotata* rhizosphere soils were analyzed, which revealed their relationship with iridoid compounds in *P. rotata* leaves. Third, 19 genes related to iridoid biosynthesis were verified by qRT-PCR, which is able to explore iridoid biosynthesis pathways. The results identified the transcripts of *P. rotata* and laid an important foundation for understanding the metabolic mechanisms of iridoid compounds in *P. rotata*.

## 2. Results

### 2.1. Analysis of Leaf Metabolome Data of P. rotata Leaves in Different Regions

Metabolites were extracted from the leaves of *P. rotata* from different regions (Figure 1; Table 1), and three replicate samples were collected from each region. Subsequently, the extracted samples underwent LC-MS/MS analysis. Principal component analysis (PCA) was used to evaluate the variability among the samples. The findings demonstrated a close proximity among the three replicated samples within the same region, whereas significant differences in the distribution of metabolites were observed between samples from different regions (Figure 2A). Furthermore, correlation analysis revealed a higher correlation within samples than between different samples, confirming the stability and reliability of the obtained metabolomic data (Figure 2B).

A total of 575 metabolites were detected, including amino acids (67), sugars and alcohols (65), terpenoids (64), alkaloids (62), organic acids (62), flavonoids (61), and others (49) (Figure 2C). Among these, 296 metabolites successfully matched with the KEGG database (Appendix A).

To further assess the variations in metabolites among the leaves of *P. rotata* from different regions, we used the criteria of FC > 1, *p* < 0.05, and VIP > 1 to screen for differentially accumulated metabolites (DAMs). The screening results were visualized using a volcano plot (Appendix A). In the clustering heatmap (Figure 2D), the accumulation of metabolites displayed a clear variation in terms of the pattern of metabolite abundance in different regions. A total of 455 DAMs were identified, with the highest number of DAMs observed in the MQ vs. YS comparison (319), whereas the MQ vs. CD comparison had the lowest number (270) (Table 2). Of these differentially abundant metabolites, 85 were common across all six datasets (Figure 2E). Additionally, CD vs. YS, HN vs. MQ, HN vs. YS, HN vs. CD, MQ vs. CD, and MQ vs. YS had five, five, six, seven, seven, and twelve unique differentially expressed metabolites, respectively (Figure 2E).

In the analysis of DAMs in *P. rotata* leaves from different regions, we observed 45 terpenoid metabolites (Appendix A). These terpenoid metabolites comprised 15 sesquiterpenes and monoterpenes, 16 diterpenoids, triterpenes, tetraterpenes, 5 triterpene saponins, and 8 iridoids. These iridoid DAMs included geniposide, shanzhiside methyl ester, agnuside, barlerin, catalpin, shanzhiside, feretoside, and 8-epideoxyloganic acid. The chemical structural formula of the eight iridoid DAMs is shown in Appendix A.

### 2.2. Overview of Transcriptome Sequencing Data of P. rotata Leaves from four Different Regions

In this study, leaves of *P. rotata* from four different regions were used for transcriptome sequencing (Figure 3). A total of 74.46 Gb of clean data were obtained from 12 libraries (four regions, three replicates), and each sample’s clean data exceeded 5.71 Gb with above 93.22% Q30 (Table 3). The GC content, with an average of 46.61%, ranged from 45.35% to 49.12% (Table 3). A total of 73,584 transcripts with an average length of 2098.50 bp and a N50 length of 2940 bp were obtained (Table 4). A total of 29,833 unigenes with an average length of 1637.21 bp and N50 length of 2786 bp were obtained (Table 4). A total of 23,540 unigenes were successfully annotated in nine public databases (NR, Pfam, KOG/NOG/COG, Swiss-Prot, KEGG, and GO), accounting for 78.91% of the unigenes, and 7099 unigenes were annotated in at least one database, accounting for 23.80% of the unigenes (Table 5). A total of 14,723 unigenes with lengths exceeding 1000 bp were obtained, accounting for 49.35% of the total unigenes (Table 5).

The PCA of the samples based on the fragments per kilobase of exon per million fragments mapped (FPKM) values showed that all biological replicates clustered together, indicating the high reliability of our results (Figure 3A). According to the results, four groups of samples were clearly distinguished in the PC1 dimension of the PCA score graph (26.14% variation) and were further differentiated in the PC2 dimension (score graph 22.02%), indicating that the gene expression varied significantly in four different regions.

A total of 15,879 DEGs (fold-change ≥ 2, false discovery rate < 0.01) were identified (Appendix A), including 13,052 DEGs that were successfully annotated and characterized and 2827 DEGs that were not annotated and uncharacterized (Figure 3B). According to the KEGG annotation results, DEGs were categorized into five groups: organic systems, metabolism, genetic information processing, environmental information processing, and cellular processes (Figure 3C). A total of 3599 DEGs were annotated in the KEGG metabolism category (Appendix A). Among the DEGs based on the KEGG database, 238 DEGs were related to the metabolism of terpenoids and polyketides (Figure 3C).

### 2.3. KEGG Enrichment of DEGs in Different Groups of Samples

The KEGG pathway enrichment analysis revealed that these DEGs were mainly enriched in several secondary metabolite metabolism pathways. The results for the top 20 KEGG pathways enriched in each group are presented as scatter plots (Figure 4). According to the enriched pathways related to terpenoid biosynthesis (Appendix A), HN vs. CD mainly included sesquiterpenoid, triterpenoid, monoterpenoid, diterpenoid, and zeatin biosynthesis (Figure 4A). CD versus MQ mainly included sesquiterpenoid and triterpenoid biosynthesis, monoterpenoid biosynthesis, and zeatin biosynthesis (Figure 4B). MQ vs. YS mainly involved sesquiterpenoid and triterpenoid biosynthesis (Figure 4C). YS and HN mainly included sesquiterpenoid and triterpenoid biosynthesis, diterpenoid biosynthesis, and carotenoid biosynthesis (Figure 4D). This analysis preliminarily identified obvious differences in the above substances in different regions and provided basic data for subsequent research.

### 2.4. Regulation of MVA/MEP/Iridoid Pathway Genes in Different Regions

Previous studies have shown that iridoids are derived from terpenoids, which are synthesized by the upstream MVA (from acetyl-CoA to IPP) and MEP (from GA-3P/pyruvate to IPP) pathways (Figure 5) [36]. These pathways produce early precursors of iridoid compounds (Figure 5) [36]. In this study, the accumulation of iridoids as active components in different regions was investigated. This study identified a large number and variety of genes in *P. rotata* that participate in the biosynthesis of iridoids. Combined with the results of previous studies, we reconstructed the iridoid biosynthesis pathway and obtained candidate unigenes from the DEGs. To obtain a systematic view of the iridoid biosynthesis pathway, we observed the abundance of 41 transcripts encoding 23 key enzyme-encoding genes involved in iridoid biosynthesis (Figure 5). The whole pathway can be divided into precursor formation, terpenoid skeleton biosynthesis, and chemical modification (Figure 5). As shown in Figure 5, 12 key enzymes were encoded by more than one unigene: *AACT* (2), *HMGR* (3), *IDI* (2), *DXS* (4), *HDS* (2), *HDR* (2), *GPPS* (2)*, 8-HGO* (3), *G10H* (2), *7-DLGT* (2), *STR* (3), and *7-DLNGT* (3). Eleven key enzymes were encoded by one unigene: *HMGS* (1), *MVK* (1), *PMK* (1), *MVD* (1), *DXR* (1), *CMS* (1), *CMK* (1), *MCS* (1), *GES* (1), *IS* (1), and *LAMT* (1) (Figure 5).

Based on the eight iridoid DAMs, we speculated upon the frame diagram of the major DAMs during the biosynthesis of iridoids in *P. rotata* (Figure 6A). Shanzhiside, geniposide, and shanzhiside methyl ester were high in HN, and feretoside, catalpin, barlerin, and shanzhiside methyl ester were high in MQ (Figure 6B). The levels of five compounds, namely agnuside, catalpin, 8-epideoxyloganic acid, geniposide, and feretoside, were all high in YS (Figure 6B). This result shows that there were five high ion abundances of iridoids in the YS region. We also performed a cluster analysis of all differentially expressed genes involved in the four pathways, and found that the changes in the expression of most genes in the MEP pathway (Figure 6D) were high. In the MEP pathway, the expression of *MCS* and *HDR2* was the highest in all four regions. In the terpenoid skeleton biosynthesis pathway (Figure 6E), *8-HGO1*, *GPPS2*, and *G10H1* expression was higher than that of other genes. As shown in Figure 6C,D, the MEP pathway was more active than the MVA pathway, indicating that the MEP pathway plays a major role in iron metabolism in *P. rotata*.

### 2.5. Analysis of Physicochemical Properties of P. rotata Rhizosphere Soil

The concentrations of five soil nutrients, namely alkali-hydrolyzable nitrogen (AN), electrical conductivity (EC), exchangeable magnesium (EM), available potassium (AK), and available phosphorus (AP), were determined. The results are shown in Figure 7A. Notable variations in the nutrient composition were observed among the different regions. AN, EC, and EM exhibited similar trends of initially increasing and then decreasing, with the highest values observed in the YS. Conversely, AK displayed a decreasing trend followed by an increase, with the lowest values observed in MQ. The AP content gradually decreased with increasing altitude.

Figure 7B illustrates the significant positive correlations between AN and EC with geniposide (*p* < 0.001). AP demonstrated a highly significant positive correlation with shanzhiside methyl ester (*p* < 0.001) and a significant negative correlation with barlerin (*p* < 0.01). Furthermore, AK exhibited a highly significant positive correlation with shanzhiside (*p* < 0.001) and a highly significant negative correlation with catalpin (*p* < 0.001).

### 2.6. Validation of DEGs Related to the Metabolism of Iridoids by qRT-PCR

To ensure the reliability and stability of the transcriptome sequencing data and gain a deeper understanding of the regulatory mechanism of iridoid biosynthesis in *P. rotata* across different regions, we selected 19 genes involved in the iridoid metabolic pathway to analyze gene expression using qRT-PCR (Figure 8A). The results demonstrated that most gene expression patterns were consistent with the findings from transcriptome analysis, except *DXR* and *HMGR2*. This further confirmed the authenticity and reliability of the transcriptome data. With increasing altitude, *DXS1*, *GPPS1*, and *HDR1* exhibited a decrease followed by an increase, and *DXS2* and *IDI1* showed a gradual increase (Figure 8A). *G10H1*, *GPPS2*, *CMK*, *HDS1*, *HMGS*, *MCS*, *MVD*, and *PMK* exhibited gradual decreases (Figure 8A). *DXR* and *8-HGO1* displayed an increase, followed by a decrease (Figure 8A). *G10H2* and *AACT1* initially increased, then decreased, and then increased again (Figure 8A). The *MVK* and *HMGR2* levels decreased, then increased, and then decreased again (Figure 8A). The correlation between the fluorescence quantitative expression of 17 genes (except *DXR* and *HMGR2* genes) and their transcripts reached a significant level (*p* < 0.05) (Appendix A).

## 3. Discussion

### 3.1. Iridoid Compounds Found in P. rotata Leaves

*P. rotata*, a common traditional Chinese medicinal herb, exhibits a broad spectrum of pharmacological activities, including anti-inflammatory, analgesic, and hepatoprotective effects [37]. Over 233 compounds have been identified from *P. rotata* through extensive research [23,24,25], indicating its rich chemical composition [38]. To further elucidate the chemical composition of *P. rotata*, this study employed a metabolomics approach to analyze the leaves of *P. rotata* collected from diverse regions. The results of our study showed that a total of 575 metabolites were detected in *P. rotata*, among which 61 belonged to flavonoids, 64 belonged to terpenoids, and 12 were iridoids. Notably, four compounds—agnuside, feretoside, asperuloside, and verbenalin—exhibited anti-inflammatory activity.

Agnuside is mainly found in *Vitex agnus-castus* [39] and *Vitex negundo* [40], and has demonstrated anti-inflammatory activity by downregulating the inflammatory mediators PGE2 and LTB4, as well as by reducing cytokine expression [41]. Feretoside is primarily found in *Eucommia ulmoides* and *Gardenia jasminoides* [42], and acts as an HSP inducer, exhibiting anti-inflammatory and cellular protective functions [43]. Asperuloside is primarily derived from *Panax ginseng* [44] and *Hedyotis diffusa*, and it inhibits inflammatory cytokines and mediators through the suppression of the NF-κB and mitogen-activated protein kinase (MAPK) signaling pathways [45]. Verbenalin is mainly present in *Verbena officinalis* [46] and exhibits hepatoprotective, anti-inflammatory, and antiviral activities [47,48]. It is hypothesized that these four iridoids components play a significant role in the anti-inflammatory action of *P. rotata*.

### 3.2. Correlation Analysis between the Contents of Iridoid Metabolites and Soil Nutrients

Soil nutrients are essential elements and compounds in the soil, and include common elements such as nitrogen, phosphorus, and potassium, as well as trace elements such as iron, zinc, and copper. Multiple studies have shown that soil nutrients play a crucial role in plant growth and development [49,50] and can influence the biosynthesis and accumulation of plant secondary metabolites [51,52]. In *Camellia sinensis*, a deficiency of soil nutrients can lead to the downregulation of the protochlorophyllide oxidoreductase-encoding gene, which is involved in chlorophyll biosynthesis, ultimately resulting in leaf yellowing [53]. In *Rosmarinus officinalis*, a significant and positive relationship was found between the concentrations of nitrogen (N) and extractable phosphorus (P_E_) and the leaf terpene content [54]. In *Linaria dalmatica*, soil nitrogen enrichment increases growth, reproduction, and whole-plant iridoid glycosides [55].

In studying *P. rotata* leaves from different regions, we found that there was a total of 455 DAMs, including eight iridoids (geniposide, shanzhiside methyl ester, agnuside, barlerin, catalpin, shanzhiside, feretoside, and 8-epideoxyloganic acid). A Pearson correlation analysis was conducted to investigate the relationship between these compounds and soil nutrients (AN, EC, EM, AK, and AP), revealing significant correlations between soil nutrients and the iridoid content. Specifically, AP showed a highly significant positive correlation with shanzhiside methyl ester (*p* < 0.001), whereas AK exhibited a highly significant positive correlation with shanzhiside (*p* < 0.001) (Figure 7B). Agnuside was significantly correlated with AN, EC, EM, and AK (Figure 7B). Therefore, it can be speculated that AP and AK play a key role in the biosynthesis of medicinal components in *P. rotata* leaves. However, the specific mechanisms of action of AP and AK in this process remain unclear, and further research is needed to understand the relationship between soil nutrients and the plant secondary metabolism.

### 3.3. Correlation Analysis between the Content of Iridoid Metabolites and the Expression of Related Genes

Transcriptomics has been used to investigate the biosynthetic pathways of iridoid compounds in various plant species, including *Tripterygium wilfordii* [56], *Catharanthus roseus* [57], and *Salvia miltiorrhiza* [58]. In the present study, 41 genes associated with iridoid metabolism were initially identified and subsequently annotated to correspond to 23 enzymes. This finding is consistent with the results of Hou et al. in *Cinnamomum camphora* [59]. We also performed a cluster analysis of all differentially expressed genes involved in the precursors of iridoids and found that expression of most genes in the MEP pathway changed significantly (Figure 6D). A correlation analysis was performed between the expression of 41 transcripts and the content of 12 iridoids (Appendix A). During the synthesis of the precursors of iridoids, the MEP pathway, which is characterized by its unique genes, exhibits 22 sets of positive correlations. In contrast, the MVA pathway, distinguished by its specific functional genes, shows 16 sets of positive correlations. Notably, the MEP pathway plays a more significant role than the MVA pathway in the synthesis of the precursors of iridoids. Kou et al. [17] also demonstrated that in *G. macrophylla*, the MEP pathway plays a predominant role in the formation of IPP.

The correlation between the expression of the 19 enzyme-encoding genes and the content of the eight iridoid metabolism-related components in *P. rotata* from different regions was analyzed (Figure 8B). Among them, there were 16 pairs of positive correlations and 23 pairs of negative correlations. Notably, shanzhiside methyl ester exhibited a significant positive correlation with the expression of eight genes (*8-HGO1*, *DXR*, *GPPS2*, *HDS1*, *PMK*, *MVD*, *MCS* and *CMK*), whereas catalpin displayed a negative correlation with the expression of eight genes (*GPPS2*, *HDS1*, *PMK*, *MVD*, *DXS1*, *GPPS1*, *G10H1* and *HMGS*) (*p* < 0.05). The mutual coordination of these genes jointly regulated the biosynthesis of the iridoid metabolism. Furthermore, *DXS1*, *IDI1*, *8-HGO1*, and *G10H2* were strongly correlated with iridoid metabolic components. The correlation between *DXS1* and shanzhiside was positive (*p* < 0.001) (Figure 8B). *DXS*, the initial rate-limiting enzyme in the MEP pathway, plays a critical role in regulating terpenoid compound biosynthesis [60,61]. Numerous studies have demonstrated that the overexpression of *DXS* leads to an increased iridoid content. For instance, Zhou et al. showed that the overexpression of the *DXS2* gene in the hairy roots of *Salvia miltiorrhiza* resulted in tanshinone accumulation [58]. *IDI1* was positively correlated with 8-epideoxyloganic acid (*p* < 0.01) and negatively correlated with shanzhiside methyl ester (*p* < 0.001) (Figure 8B). *IDI*, which participates in both the MVA and MEP pathways, plays a crucial role in catalyzing the interconversion of IPP and DMAPP, thereby regulating their ratio and influencing the synthesis of downstream products [62]. The overexpression of *IDI* expedites the accumulation of β-carotene in *Escherichia coli* transformants [63]. The correlation between *G10H2* and barlerin was positive (*p* < 0.001) (Figure 8B). G10H, a cytochrome P450 monooxygenase, is involved in the biosynthesis of iridoid compounds and several monoterpenoid alkaloids found in various plants [64]. The overexpression of *G10H* in *Swertia mussotii* led to simultaneous increases in the contents of 10-hydroxygeraniol and swertiamarin [65]. *8-HGO1* was negatively correlated with 8-epideoxyloganic acid (*p* < 0.001) and positively correlated with shanzhiside methyl ester (*p* < 0.01) (Figure 8B). *8-HGO* plays a key role in the secoiridoid pathway and catalyzes secologanin synthesis [66]. Under UV-B stress, the expression of the *10-HGO* gene in the leaves of *Catharanthus roseus* can upregulate and increase the content of its downstream product, strictosidine [67]. In our study, *DXS1*, *IDI1*, *8-HGO1*, and *G10H2* were found to regulate iridoid biosynthesis in *P. rotata*.

## 4. Materials and Methods

### 4.1. Plant Materials

Seedlings of *P. rotata* were collected from four different areas of Qinghai Province, with altitudes ranging from 3540 m to 4270 m during the flowering period (Figure 1 and Figure 9, Table 1). This plant mainly grows in the soil of highland meadows, and is usually accompanied by plants such as *Saussurea hieracioides*, *Argentina anserina*, *Nasturtium officinale*, *Dontostemon pinnatifidus*, *Lancea tibetica*, *Gentiana crassicaulis*, *Juncus prismatocarpus*. The collection sites included Henan County (HN) at an altitude of 3540 m, Maqin County (MQ) at an altitude of 3750 m, Yushu City (YS) at an altitude of 3880 m, and Chengduo County (CD) at an altitude of 4270 m. Only healthy *P. rotata* plants of similar size were selected for the study. In the study, 10–15 healthy plants with consistent growth were randomly selected from each area. The entire plant was carefully excavated and washed multiple times with water to remove the topsoil. Subsequently, the plants were sterilized with 75% ethanol for 5 min and rinsed twice with sterile water. The leaves were used as experimental materials and were immediately frozen in liquid nitrogen. The frozen samples were stored in a freezer at −80 °C for subsequent transcriptome and metabolomic analyses.

### 4.2. Preparation of Samples from the Leaves of P. rotata

Samples were extracted from the leaves of *P. rotata* from different regions with three biological replicates. First, the samples were vacuum freeze-dried and weighed to 50 mg, and 1000 μL of extraction solution (methanol:acetonitrile:water = 2:1:1) was added. Subsequently, the sample was ground using a grinder (45 Hz, 10 min) and sonicated in an ice water bath for 10 min. Afterward, the sample was placed in a −20 °C refrigerator for 1 h. The stationary samples were placed in a centrifuge and centrifuged at 4 °C and 12,000 rpm for 10 min. The supernatant (500 μL) was collected and dried using a vacuum concentrator, and 160 μL of 50% acetonitrile solution was added to dissolve the dried extract. The dissolved samples were mixed by vortex mixing, placed again in an ice water bath for 10 min, and centrifuged again at 4 °C and 12,000 rpm for 10 min. The supernatant (120 μL) was collected and stored in a 2 mL injection bottle, with 10 μL from each sample mixed to create a QC sample for subsequent machine detection.

### 4.3. LC-MS/MS Analysis

LC-MS/MS analysis was performed using a Whatsch Acquisition I-Class PLUS ultrahigh-performance liquid (Waters, Milford, MA, USA) chromatography-tandem AB Scienx Qtrap 6500+ high-sensitivity mass spectrometer (Sciex, Framingham, MA, USA). The conditions for the ultrahigh-performance liquid chromatography were as follows: the chromatographic column was a Waters Acquisition UPLC HSS-T3 (1.8 µm, 2.1 mm × 100 mm). Mobile Phase A was ultrapure water (containing 0.1% formic acid and 5 mM ammonium acetate), and Phase B was acetonitrile (with 0.1% formic acid added). The gradient elution program was used for sample measurement, with initial conditions of 98% A and 2% B maintained for 1.5 min, and these conditions then adjusted to 50% A and 50% B within 5 min. The linear gradient was adjusted to 2% A and 98% B within 9 min, maintained for 1 min, adjusted to 98% A and 2% B within 1 min, and maintained for 3 min. The flow rate was set to 350 μL/min, and the column temperature was set to 50 °C. The effluent was alternatively connected to an ESI-triple quadrupole linear ion trap (QTRAP)-MS.

The ESI source operation parameters were as follows: ion source temperature, 550 °C; ionizing spray voltage (IS), 5500 V (positive ion mode)/−4500 V (negative ion mode). The ionization source gas I (GSI), gas II (GSII), and curtain gas (CUR) were set at 50, 55, and 35 psi, respectively; the collision-activated dissociation (CAD) was medium. Instrument tuning and mass calibration were performed with 10 and 100 μmol/L polypropylene glycol solutions in the QQQ and LIT modes, respectively. QQQ scans were acquired as MRM experiments, with the collision gas (nitrogen) set to medium. The DP (declustering potential) and CE (collision energy) for individual MRM transitions was evaluated with further DP and CE optimization. A specific set of MRM transitions was monitored for each period according to the metabolites eluted within this period.

### 4.4. Qualitative and Quantitative Analysis of Metabolites

Metabolite mass spectrometry analysis data for different samples were obtained using Analyst 1.6.3. The peak areas of all mass spectral peaks of the substances were integrated, and the relative content of each component was calculated using the peak area normalization method. An internal standard was used for data QC (reproducibility). Samples with metabolite features and a relative standard deviation (RSD) of QC > 30% were discarded. The identified compounds were searched for classification and pathway information in the KEGG, HMDB and lipidmaps databases. According to the grouping information, which calculated and compared the difference multiples, the T test was used to calculate the difference significance pvalue of each compound. The R language package ropls (v 1.34.0) was used to perform OPLS-DA modeling, and 200 times permutation tests were performed to verify the reliability of the model. The VIP value of the model was calculated using multiple cross-validation. The method of combining the difference multiple, the P value and the VIP value of the OPLS-DA model was adopted to screen the differential metabolites. The screening criteria were FC > 1, *p* value < 0.05 and VIP > 1. The difference metabolites of the KEGG pathway enrichment significance were calculated using the hypergeometric distribution test.

### 4.5. Transcriptome Sequencing and Data Analyses

Total RNA extraction from different regions of *P. rotata* leaves was performed using TRIzol^®^ reagent (Invitrogen, St. Louis, MO, USA) following the manufacturer’s instructions. The RNA concentration, purity, and integrity were assessed using Nanodrop (Thermo Scientific, St. Louis, MO, USA) and Agilent Bioanalyzer 2100 (Agilent Technologies Inc., Santa Clara, CA, USA). For each group, 1 μg of RNA was used to construct a sequencing library using the NEBNext^®^ Ultra™ RNA Library Prep Kit for Illumina^®^ (NEB, Ipswich, MA, USA). The resulting double-stranded cDNA fragments were size-selected using the AMPure XP system (Beckman Coulter, Beverly, MA, USA), followed by PCR enrichment to generate a cDNA library. Subsequently, the Illumina NovaSeq 6000 sequencing platform was used for PE150 mode (2 × 150 bp) sequencing. The sequencing data were further analyzed using the BMKCloud online platform (www.biocloud.net/, accessed on 10 August 2023).

A custom Perl script was used to process the raw data (raw reads) in the Fastq format, resulting in clean data (clean reads), and the Q20, Q30, and GC contents of the clean data were calculated. Subsequently, Trinity software (v 2.14.0) was used to assemble the clean data obtained from *P. rotata*. The assembly parameters were set as follows: min_kmer_cov = 2. The other parameters were set to default values. All the assembled transcripts were searched using BLASTX against the sequence information within the NCBI protein non-redundant (NR), COG, and KEGG databases to identify the proteins whose sequences were most similar to those of the given transcripts to retrieve their functional annotations, and a typical cutoff E-value of less than 1.0 × 10^−5^ was set. The BLAST2GO (http://www.blast2go.com/b2ghome, accessed on 11 October 2023) [68] program was used to obtain GO annotations of the unique assembled transcripts to describe the biological processes, molecular functions, and cellular components. Metabolic pathway analysis was performed using the KEGG database (http://www.genome.jp/kegg/, accessed on 18 October 2023) [69].

The gene expression levels in each sample were assessed using RSEM. The fragments per kilobase of script per million mapped reads (FPKM) value was used to quantify the expression abundance of each unigene. Differential expression analyses were performed using fold change ≥ 2 and false discovery rate < 0.01 as the standard, using the DESeq2 software (v 1.39.0) package. Furthermore, the KEGG enrichment analysis was conducted using clusterProfiler software (v 4.4.4) to assess the degree of enrichment of DEGs within the KEGG pathway.

### 4.6. Soil Sample Collection and Physicochemical Property Measurements

Soil samples were collected from the rhizosphere of *P. rotata* plants, and four sampling points were chosen. The collected soil samples were manually homogenized to ensure the removal of roots and coarse plant debris to obtain a composite sample. Subsequently, the composite sample was stored in a well-ventilated and shaded area for drying, allowing the assessment of soil physical and chemical properties.

Available phosphorus (AP) in the soil was extracted using the sodium bicarbonate molybdenum antimony resistance colorimetry method. Alkali-hydrolyzable nitrogen (AN) was quantified using conductometric titration [70]. The available potassium (AK) content was measured using the NH4OAc extraction–flame photometer method [71]. Additionally, the exchangeable calcium (EC) and magnesium (EM) content in the soil was determined using atomic absorption spectrometry after NH4OAc extraction.

### 4.7. qRT PCR Analysis

Based on the transcriptome sequences of *P. rotata* from four different regions, we screened the sole gene sequences of nineteen genes (*DXS1*, *DXS2*, *G10H1*, *G10H2*, *GPPS1*, *GPPS2*, *AACT1*, *CMK*, *DXR*, *HDR1*, *HDS1*, *HMGS*, *IDI1*, *MCS*, *MVK*, *MVD*, *PMK*, *HMGR2*, and *8-HGO1*) involved in the metabolism of iridoids in *P. rotata* (Appendix A). The *CYP22* gene was selected as an internal reference based on a previous screening study [72]. Online Primer Premier software (version 5.0) was used to design the gene-specific primers synthesized by Aoke Dingsheng Biotechnology Co., Ltd. (Xian, China). The primers used are listed in Appendix A.

Total RNA was extracted from the leaves of *P. rotata* using a FastPure^®^ Plant Total RNA Isolation Kit (Polysaccharides & Polyphenolics-rich) (Nanjing Novozan Biotechnology Co., Ltd., Nanjing, China). The integrity of the RNA was detected by 1.2% agarose gel electrophoresis, and the RNA concentration was determined using Nanodrop 2000 (Thermo Scientific, St. Louis, MO, USA). First-strand cDNA was synthesized using a reverse transcription kit (Hiscript^®^ qRT-PCR Supermax) (Nanjing Novozan Biotechnology Co., Ltd., Nanjing, China), and the reverse transcription products were stored at −20 °C.

qRT-PCR was performed using first-strand cDNA synthesized as a template with the ChamQ Universal SYBR qPCR Master Mix Kit. qRT-PCR system: 10 μL of 2× ChamQ Universal SYBR qPCR Master Mix, 7.2 μL of ddH2O, 0.4 μL of upstream and downstream primers (Appendix A), and 2 μL of cDNA, for a total of 20 μL. The reaction procedure was as follows: denaturation at 95 °C for 30 s; cycling at 95 °C for 5 s, 60 °C for 30 s, and 40 cycles; and melting at 95 °C for 15 s, 60 °C for 50 s, and 95 °C for 15 s. The negative control and three replicates per group were set, and the reaction was carried out on a QuantStudio 6 Flex PCR instrument. qRT-PCR was performed using the 2^−ΔΔCt^ method to analyze the relative expression of the genes to be tested [73].

### 4.8. Data Statistics and Analysis

All data in this study were analyzed based on three independent biological replicates: RNA sequencing, LC-MS/MS, and qRT-PCR analysis. The principal component analysis (PCA) of metabolites was performed using prcomp (R base function) 3.6.1. The prcomp package was used for the analysis, and factoextra was used for ggplot2-based visualization. The abscissa of the PCA score chart represents the first principal component, PC1, and the ordinate represents the second principal component, PC2. Statistical analysis was conducted using SPSS 20 (version 21; IBM, Armonk, NY, USA), and the results are expressed as the mean ± SEM. Statistical analysis was conducted using one-way ANOVA followed by Dunnett’s post hoc test. The correlation heat map was drawn using OmicShare tools (https://www.omicshare.com/, accessed on 23 December 2023).

## 5. Conclusions

In conclusion, transcriptome and metabolite analyses of *P. rotata* leaves have been performed in different habitats. The results showed eight iridoid DAMs in *P. rotata* leaves, and 19 genes related to iridoid metabolism were used for qRT-PCR analysis. Furthermore, the AP and AK in *P. rotata* rhizosphere soils shed light on the iridoid biosynthesis that occurs in *P. rotata* leaves. *DXS1*, *IDI1*, *8-HGO1*, and *G10H2* regulate iridoid biosynthesis in *P. rotata*. These results provide a broader and better understanding of the metabolic processes of iridoids from the Tibetan medicinal plant *P. rotata*.

## Figures and Tables

**Figure 1 plants-13-01627-f001:**
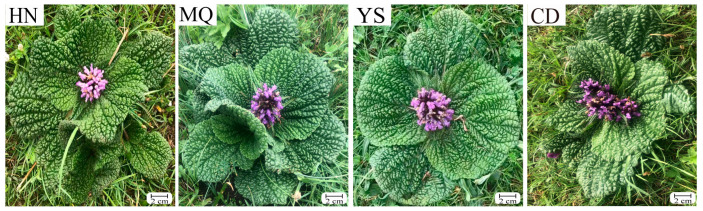
*P. rotata* in four different altitudes of Qinghai–Tibet Plateau. HN: Henan county in Qinghai province; MQ: Maqin county in Qinghai province; YS: Yushu city in Qinghai province; CD: Chengduo county in Qinghai province. Bar = 2 cm.

**Figure 2 plants-13-01627-f002:**
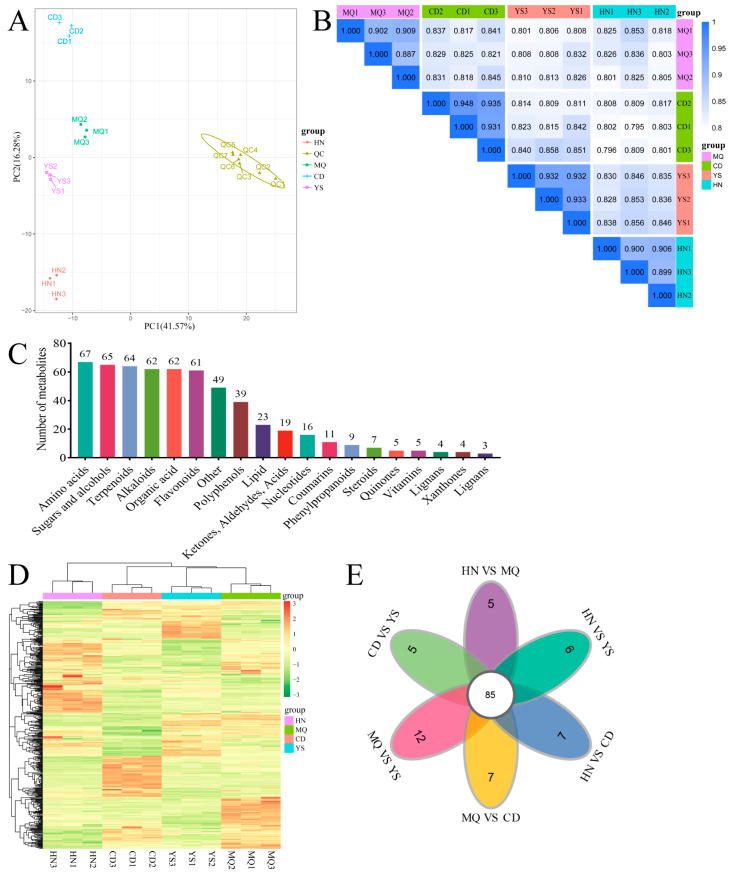
Metabonomic data analysis in leaves of *P. rotata* from four different altitudes. HN, MQ, YS, and CD from four different altitude regions. (**A**) The principal component analysis. (**B**) The correlation analysis. (**C**) Classification of metabolites. (**D**) DAMs clustering heat map. (**E**) DAM petal diagrams analyzed by OmicShare tools (https://www.omicshare.com/, accessed on 21 December 2023).

**Figure 3 plants-13-01627-f003:**
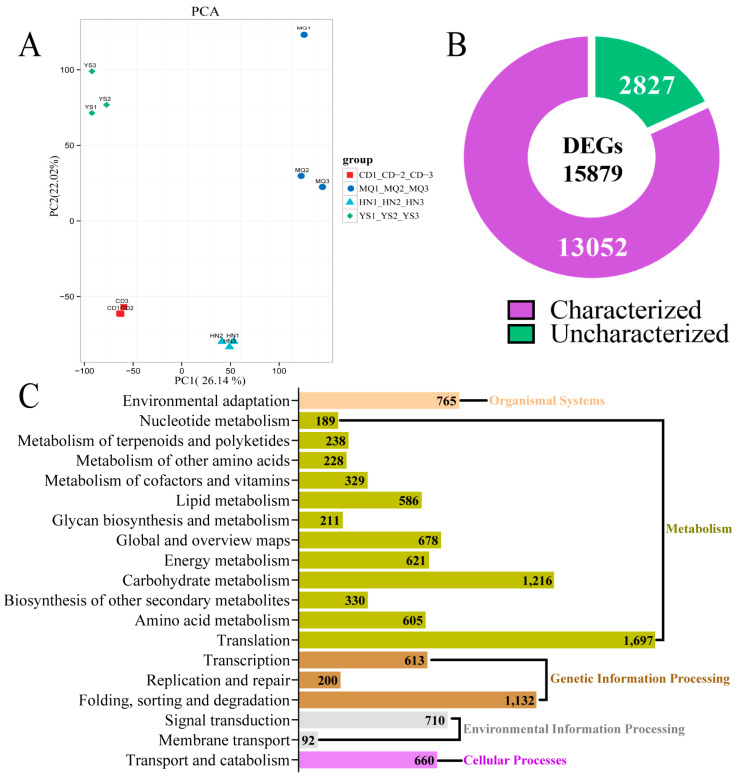
Transcriptome data analysis in leaves of *P. rotata* from four different altitudes. (**A**) Principal component analysis based on the gene expression profiles. (**B**) Annotation of DEGs. (**C**) Classification of DEGs based on the KEGG database.

**Figure 4 plants-13-01627-f004:**
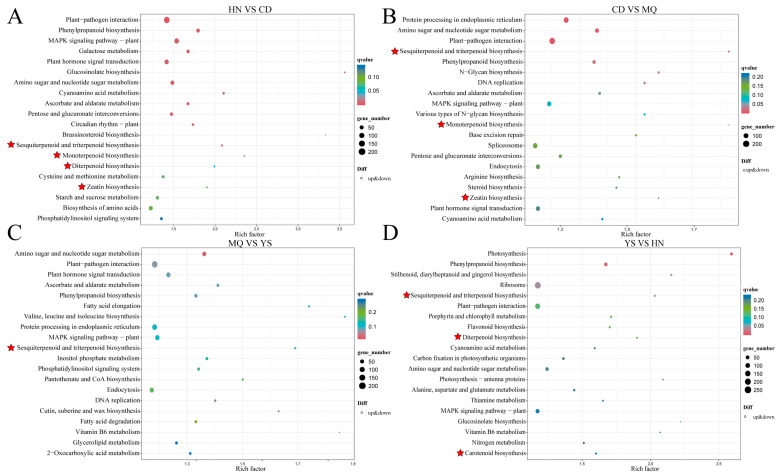
Top 20 KEGG enrichment pathways for DEGs between (**A**) HN vs. CD, (**B**) CD vs. MQ, (**C**) MQ vs. YS, (**D**) YS vs. HN. The dot color represents the qvalue, and the dot size represents the number of DEGs. The red star denotes the metabolic pathway related to the synthesis of iridoids in *P. rotata*.

**Figure 5 plants-13-01627-f005:**
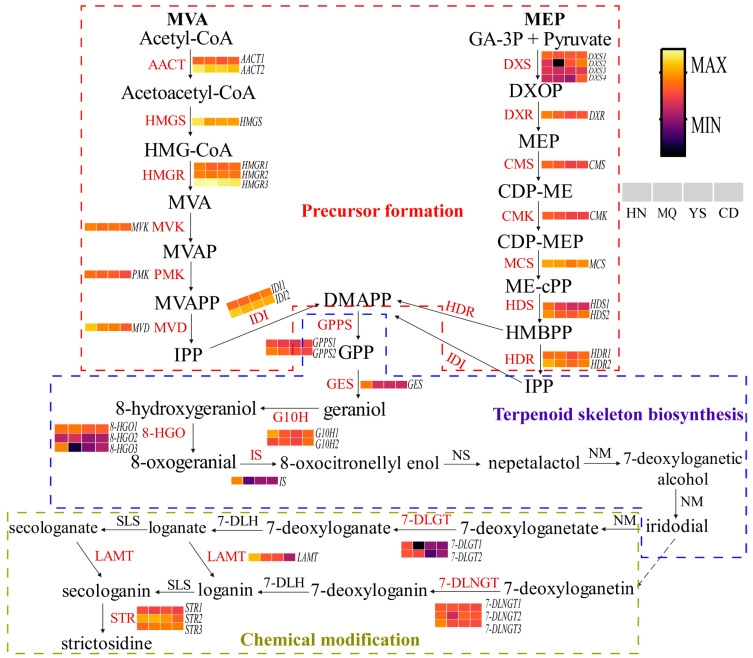
Expression analysis of genes involved in iridoid biosynthesis. Different color blocks represent the gene expression levels in four different regions of *P. rotata*. The blocks from left to right represent HN, MQ, YS, and CD from four different altitude regions. Yellow: higher FPKM; Black: lower FPKM.

**Figure 6 plants-13-01627-f006:**
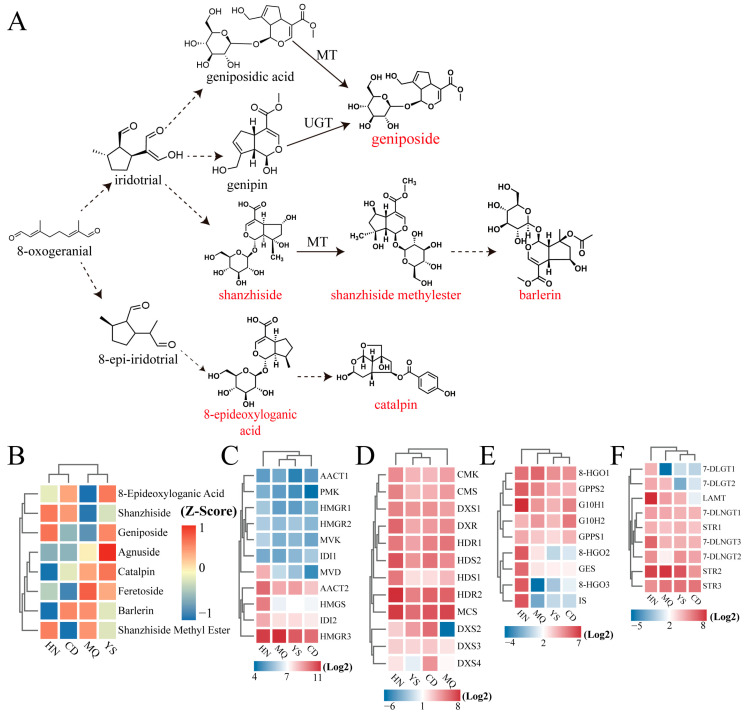
Heatmap and hierarchical clustering showing differentially expressed genes (DEGs) in four pathways in *P. rotata* leaves. (**A**) Frame diagram of major DAMs during biosynthesis of iridoids; (**B**) The ion abundance of iridoids in *P*. *rotata*; (**C**) DEGs involved in MVA pathway; (**D**) DEGs involved in MEP pathway; (**E**) DEGs involved in terpenoid skeleton biosynthesis pathway; (**F**) DEGs involved in chemical modification pathway; HN, MQ, YS, and CD from four different altitude regions. Rectangles marked with a red (deep red) and blue background represent the increased and reduced expression of genes, respectively.

**Figure 7 plants-13-01627-f007:**
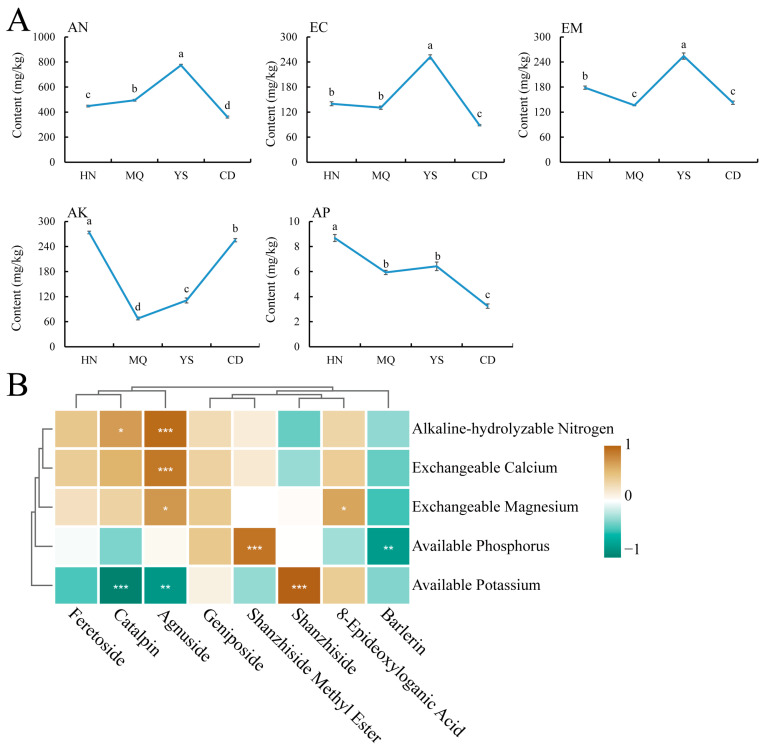
Soil nutrient analysis. HN, MQ, YS, and CD from four different altitude regions. (**A**) Changes in soil nutrient content of *P. rotata* in different regions. Lowercase letters indicate significant differences between groups (*p* < 0.05) (**B**) Correlation analysis between soil nutrients and eight compounds related to iridoid biosynthesis, respectively. Brown and blue represent positive and negative correlations. *: 0.01 < *p* < 0.05; **: 0.001 < *p* < 0.01; ***: *p* ≤ 0.001.

**Figure 8 plants-13-01627-f008:**
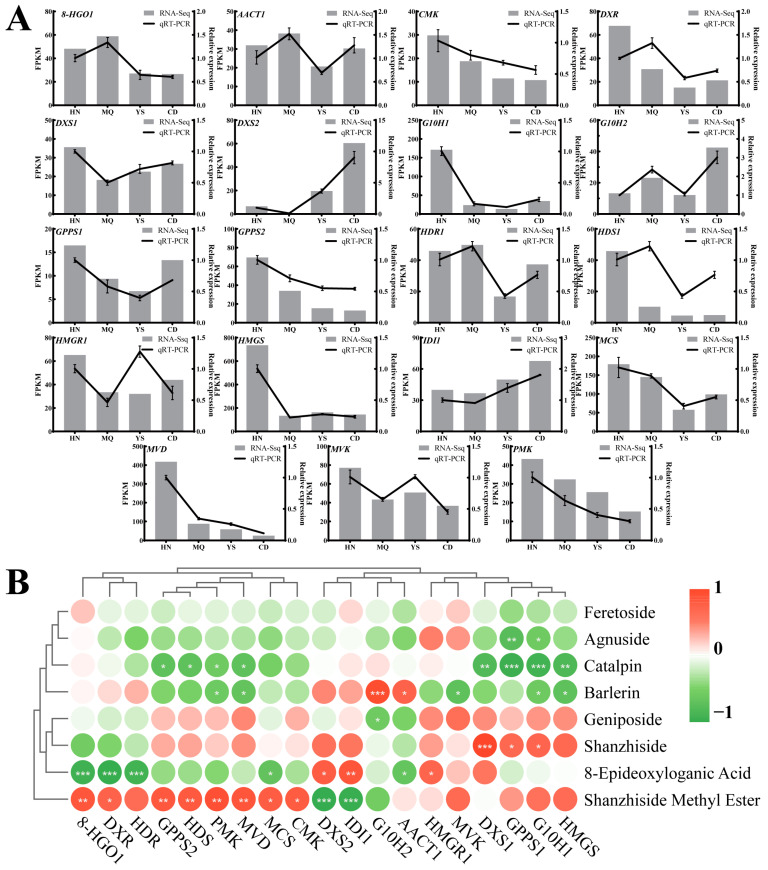
Transcripts and qRT-PCR expression analysis in leaves of *P. rotata*. HN, MQ, YS, and CD from four different altitude regions. (**A**) Validation of transcriptomic data by qRT-PCR analysis. (**B**) Correlation analysis between qRT-PCR and eight compounds related to iridoid biosynthesis, respectively. Red and blue represent positive and negative correlations. *: 0.01 < *p* < 0.05; **: 0.001 < *p* < 0.01; ***: *p* ≤ 0.001.

**Figure 9 plants-13-01627-f009:**
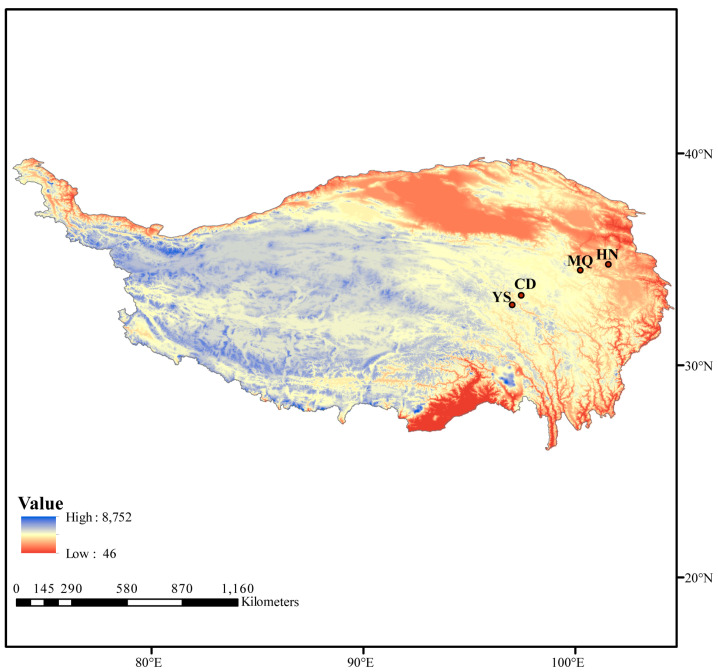
*P. rotata* sampling geographic location information.

**Table 1 plants-13-01627-t001:** Sampling location of *Phlomoides rotata* from four different regions.

Sample ID	Sampling Location	Altitude	East Longitude	North Latitude
HN	Henan County, Qinghai Province	3540 m	101°33′36″	34°45′36″
MQ	Maqin County, Qinghai Province	3750 m	100°14′38″	34°29′10″
YS	Yushu City, Qinghai Province	3880 m	97°1′23″	32°51′4″
CD	Chengduo County, Qinghai Province	4270 m	97°27′16″	33°18′2″

**Table 2 plants-13-01627-t002:** The number of DAMs in *P. rotata* leaves.

Group	DAM Numbers	Up Number	Down Number
CD vs. YS	299	158	141
HN vs. MQ	297	174	123
HN vs. YS	282	148	134
HN vs. CD	299	144	155
MQ vs. CD	270	104	166
MQ vs. YS	319	132	187

**Table 3 plants-13-01627-t003:** Transcriptomic sequencing yield quality evaluation of *P. rotata* leaves.

Sample	Clean Reads (M)	Clean Bases (G)	GC Content (%)	Q30 (%)
CD1	20.82	6.22	46.27	94.13
CD2	20.61	6.16	46.32	94.95
CD3	21.15	6.32	46.22	94.53
MQ1	20.43	6.10	49.12	93.59
MQ2	23.18	6.92	47.32	95.11
MQ3	19.26	5.75	46.72	95.11
HN1	22.16	6.63	45.44	95.72
HN2	20.58	6.16	45.40	95.25
HN3	19.10	5.71	45.35	94.98
YS1	21.26	6.28	46.13	93.22
YS2	21.39	6.32	47.20	94.23
YS3	19.80	5.90	47.82	94.31

Note: Each sample has three replications.

**Table 4 plants-13-01627-t004:** Evaluation of transcriptome-optimized assembly results of *P. rotata*.

	Transcript	Unigene
Total Number	73,584	29,833
Total Length	154,415,732	48,842,910
300–500 (bp)	12,718	8543
500–1000 (bp)	11,026	5933
1000–2000 (bp)	11,409	5025
2000+ (bp)	38,431	10,332
Mean Length	2098.50	1637.21
N50 Length	2940	2786

**Table 5 plants-13-01627-t005:** Annotation statistics for all unigenes of *P. rotata* leaves.

Database Type	Number of Unigenes	Length ≥ 1000
COG	7099	5111
GO	18,406	12,329
KEGG	15,207	10,501
KOG	12,901	8894
Pfam	17,074	12,143
Swissprot	14,647	10,377
TrEMBL	21,454	14,062
eggNOG	17,911	12,236
NR	23,041	14,625
Total number of Unigenes	23,540	14,723

## Data Availability

The raw RNA-seq datasets can be found in the NCBI SRA under the project number: PRJNA1076649. https://www.ncbi.nlm.nih.gov/sra/PRJNA1076649/, accessed on 15 March 2023.

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
