# Peer review of "A Transcriptomic and Metabolomic Study on the Biosynthesis of Iridoids in *Phlomoides rotata* from the Qinghai–Tibet Plateau"

_plants, 2024, doi:10.3390/plants13121627_

Round 1

Reviewer 1 Report

Comments and Suggestions for Authors

In the manuscript titled ‘Transcriptomic and Metabolomic Study on the Biosynthesis of Iridoid in Phlomoides rotata from the Qinghai-Tibet Plateau,’ the authors performed transcriptomic and metabolomic analyses on P. rotata leaves from different regions in China. They found differentially accumulated metabolites in iridoid metabolism. The authors also correlated this information with the physicochemical properties of P. rotata soil.

Comments:

1)      The authors should consider revising their title to ‘A Transcriptomic and Metabolomic Study on the Biosynthesis of Iridoids in P. rotata from the Qinghai-Tibet Plateau.’

22)    Regarding the identification of the four new metabolites, can the authors provide LC-MS or LC-MS/MS chromatograms for each compound together with their corresponding MS/MS spectra? I did not observe this in the manuscript. If chemical standards are available or metabolite information is published, the authors could make their identifications based on known LC retention time and MS/MS spectra. Otherwise, putative identification via LC-MS/MS can be used.

3)     Can the authors speculate on why the MEP pathway appears to play an important role in iridoid metabolism compared to the MVA pathway in P. rotata?

4)      In addition, can the authors speculate on the significance of the negative correlations between gene transcripts and iridoid metabolites (e.g., those described in fig. 8)?

5)     Were the authors able to generate MVA and MEP pathway data via LC-MS/MS analysis? If so, this data could provide further evidence of the extent of flux through both pathways.

Comments on the Quality of English Language

Comment:

1) The manuscript is very well written. I only have minor comments regarding the title and figure 7 (see below).

2) On the Fig. 7 title, can the authors please change to ‘Soil nutrient analysis.’

Author Response

Q1: The authors should consider revising their title to ‘A Transcriptomic and Metabolomic Study on the Biosynthesis of Iridoids in P. rotata from the Qinghai-Tibet Plateau.

R1: We have changed the title in to ‘A Transcriptomic and Metabolomic Study on the Biosynthesis of Iridoids in Phlomoides rotata from the Qinghai-Tibet Plateau.’

Q2: Regarding the identification of the four new metabolites, can the authors provide LC-MS or LC-MS/MS chromatograms for each compound together with their corresponding MS/MS spectra? I did not observe this in the manuscript. If chemical standards are available or metabolite information is published, the authors could make their identifications based on known LC retention time and MS/MS spectra. Otherwise, putative identification via LC-MS/MS can be used.

R2: The metabolomics analysis for this study was conducted by Biomarker Technology Company (Shanghai, China), which does not provide mass spectrometry data due to commercial confidentiality. Consequently, to ensure the rigor of the paper, we have removed the claim about the four newly identified compounds.

Q3: Can the authors speculate on why the MEP pathway appears to play an important role in iridoid metabolism compared to the MVA pathway in P. rotata?

R3: The MEP pathway exhibits significant changes in gene expression. As shown in Figure S4 (Correlation between transcripts related to the metabolism of iridoids and 12iridoids), the MEP pathway has more correlations with iridoid components compared to the MVA pathway. This suggests that the MEP pathway plays a more crucial role in iridoid metabolism in P. rotata.

Q4: In addition, can the authors speculate on the significance of the negative correlations between gene transcripts and iridoid metabolites (e.g., those described in fig. 8)?

R4: The description of negative correlations has been added in lines 370-372.

Q5: Were the authors able to generate MVA and MEP pathway data via LC-MS/MS analysis? If so, this data could provide further evidence of the extent of flux through both pathways.

R5: We performed correlation analysis between transcript expression related to iridoid metabolism and iridoid metabolite content. The specifics of this analysis are provided in lines 357-362 and illustrated in Figure S4.

Reviewer 2 Report

Comments and Suggestions for Authors

In this manuscript, the authors presented transcriptomic and metabolomic studies on the biosynthesis of iridoids in Phlomoides rotate from the Qinghai-Tibet plateau. The manuscript is well written, and the results are clearly presented. The manuscript has a merit to be published in Plants. To make this manuscript even better, please consider the following comments.

1.    Line 72; The authors should add more detailed information regarding the chemical constituents reported so far from Phlomoides rotata. Also, evidence should be provided that the main constituents of Phlomoides rotata are iridoids.

2.    Line 93; The authors collected Phlomoides rotata at four different locations and altitudes in the Qinghai-Tibet plateau. The authors should add a map showing where the four plants were collected. The authors should also add information regarding the identification of the four plants mentioned above. In this regard, the authors should consider the ITS region of the leaf DNA of the four plants mentioned above. Is there any possibility that the four plants are hybrids? Comments should also be made on the vegetation at the collection sites for each of the four plants.

3.    Line 106; The authors should add the structural formulas of four newly found iridoids from Phlomoides rotata. Also, information should be added that provides evidence that the four iridoids mentioned above were first identified

4.    Line 122; The authors should add the structural formulas of eight iridoid DAMs to the manuscript.

5.    Lines 208 and 226; For the compounds listed in Figures 5 and 6, the structural formulas should also be listed along with their names. That would be more beneficial to readers of this journal.

6.    Line 275; The authors report the identification of four new iridoids from Phlomoides rotata. However, there is no detailed information regarding the structure determination of the four iridoids mentioned above. The authors should identify these iridoids using NMR and MS spectra.

Author Response

Q1: Line 72; The authors should add more detailed information regarding the chemical constituents reported so far from Phlomoides rotata. Also, evidence should be provided that the main constituents of Phlomoides rotata are iridoids.

R1: We have added a detailed description of the chemical constituents of Phlomoides rotata in lines 78-86. Additionally, we have included references to support the claim that iridoids are the main constituents of Phlomoides rotata.

Q2: Line 93; The authors collected Phlomoides rotata at four different locations and altitudes in the Qinghai-Tibet plateau. The authors should add a map showing where the four plants were collected. The authors should also add information regarding the identification of the four plants mentioned above. In this regard, the authors should consider the ITS region of the leaf DNA of the four plants mentioned above. Is there any possibility that the four plants are hybrids? Comments should also be made on the vegetation at the collection sites for each of the four plants.

R2: We have added Figure 9 in line 411 to show the geographic locations where the four Phlomoides rotata plants were collected. Because our study does not focus on plant taxonomy, we did not explore the ITS region of the leaf DNA of the four plants. We also have not performed molecular marker identification to determine if the plants are hybrids. Information regarding the vegetation at the collection sites for each of the four plants has been added in lines 399-401.

Q3: Line 106; The authors should add the structural formulas of four newly found iridoids from Phlomoides rotata. Also, information should be added that provides evidence that the four iridoids mentioned above were first identified.

R3: The metabolomics analysis for this study was conducted by Biomarker Technology Company (Shanghai, China), which does not provide mass spectrometry data due to commercial confidentiality. Consequently, to ensure the rigor of the paper, we have removed the claim about the four newly identified compounds. We have included the chemical structures of the iridoids, which are now shown in Figure S2.

Q4: Line 122; The authors should add the structural formulas of eight iridoid DAMs to the manuscript.

R4: We have added the chemical structures of the eight iridoid DAMs to the manuscript. These can be found in Line 128 and Figure S2.

Q5: Lines 208 and 226; For the compounds listed in Figures 5 and 6, the structural formulas should also be listed along with their names. That would be more beneficial to readers of this journal.

R5: Given that Figure 6A depicts differential metabolites of iridoids screened from different regions, which is a crucial result of the article, we have updated Figure 6A to include the chemical structures alongside the names of the compounds. However, as many components in Figure 5 are not the primary focus of this study, their chemical structures have not been included. If deemed necessary, I will consider adding them in future revisions.

Q6: Line 275; The authors report the identification of four new iridoids from Phlomoides rotata. However, there is no detailed information regarding the structure determination of the four iridoids mentioned above. The authors should identify these iridoids using NMR and MS spectra.

R6: The metabolomics analysis for this study was conducted by Biomarker Technology Company (Shanghai, China), which does not provide mass spectrometry data due to commercial confidentiality. Consequently, to ensure the rigor of the paper, we have removed the claim about the four newly identified compounds.

Round 2

Reviewer 2 Report

Comments and Suggestions for Authors

This revised manuscript has been modified according to the reviewer’s comments. It is acceptable for publication.